# Porous Carbon Spheres Derived from Hemicelluloses for Supercapacitor Application

**DOI:** 10.3390/ijms23137101

**Published:** 2022-06-26

**Authors:** Yuanyuan Wang, Chengshuai Lu, Xuefei Cao, Qiang Wang, Guihua Yang, Jiachuan Chen

**Affiliations:** 1State Key Laboratory of Biobased Material and Green Papermaking, Qilu University of Technology, Shandong Academy of Sciences, Jinan 250353, China; wyy1989zyc@163.com (Y.W.); lcs011129@163.com (C.L.); wangqiang8303@163.com (Q.W.); ygh2626@126.com (G.Y.); 2Beijing Key Laboratory of Lignocellulosic Chemistry, Beijing Forestry University, Beijing 100083, China

**Keywords:** hemicelluloses, carbon microspheres, hydrothermal carbonization, supercapacitor

## Abstract

With the increasing demand for dissolving pulp, large quantities of hemicelluloses were generated and abandoned. These hemicelluloses are very promising biomass resources for preparing carbon spheres. However, the pore structures of the carbon spheres obtained from biomass are usually poor, which extensively limits their utilization. Herein, the carbon microspheres derived from hemicelluloses were prepared using hydrothermal carbonization and further activated with different activators (KOH, K_2_CO_3_, Na_2_CO_3_, and ZnCl_2_) to improve their electrochemical performance as supercapacitors. After activation, the specific surface areas of these carbon spheres were improved significantly, which were in the order of ZnCl_2_ > K_2_CO_3_ > KOH > Na_2_CO_3_. The carbon spheres with high surface area of 2025 m^2^/g and remarkable pore volume of 1.07 cm^3^/g were achieved, as the carbon spheres were activated by ZnCl_2_. The supercapacitor electrode fabricated from the ZnCl_2_-activated carbon spheres demonstrated high specific capacitance of 218 F/g at 0.2 A/g in 6 M KOH in a three-electrode system. A symmetric supercapacitor was assembled in 2 M Li_2_SO_4_ electrolyte, and the carbon spheres activated by ZnCl_2_ showed excellent electrochemical performance with high specific capacitance (137 F/g at 0.5 A/g), energy densities (15.4 Wh/kg), and good cyclic stability (95% capacitance retention over 2000 cycles).

## 1. Introduction

In recent years, with the increasing consumption of petrochemical resources and the deterioration in environmental pollution, more and more attention has been paid to the development of renewable and sustainable energy storage devices. The main application power sources include supercapacitors, fuel cells, lithium-ion batteries, etc., among which, supercapacitors are one of the most promising electrochemical energy storage devices due to their high power density and short charging and discharging time [1,2]. Supercapacitors can be divided into pseudocapacitors and electric double-layer capacitors (EDLC) according to the charge storage mechanisms. EDLC have many advantages, such as superior stability and long cycle life, which make them attractive power sources for emergency power supply systems, hybrid cars, portable electronic devices, etc. [3,4]. However, the low energy density of supercapacitors (<10 Wh/kg) limits their practical application [5]. In addition, the application scope and electrochemical performance of the supercapacitors are influenced by the price and properties of the electrode materials to a certain extent. Therefore, more and more researchers have concentrated on improving the properties of electrode materials to enhance the electrochemical performance of supercapacitors [6,7,8].

Carbon materials, such as carbon fiber, graphene, and carbon microspheres, have the advantages of strong chemical stability and good electrical conductivity, making them a good choice for electrode materials in supercapacitors [9,10,11,12]. Numerous works found that electrode materials with large specific surface area and abundant pore structure can increase the storage of the electrolyte and reduce the transmission resistance of electrolyte ions, which is beneficial for improving the electrochemical performance of supercapacitors [13,14,15,16]. Porous carbon spheres are usually considered as preferable electrode materials due to their regular shape, large specific surface area, low surface free energy, and high packing density [17,18]. The porous carbon spheres are mainly prepared from nonrenewable and expensive petroleum-based resources, such as styrene, toluene, benzene, pitch, etc. [19]. Recently, hydrothermal carbonization of biomass has been one of the most attractive routes for the preparation of carbon, since this method is green, simple, and eco-friendly [18,20]. Among various biomass resources for preparing carbon spheres, hemicelluloses are very promising because they are considered as the second most abundant renewable resource only after cellulose. Furthermore, they are cheap, easy to extract, and usually treated as by-products of pulp and paper industry. However, the pore structures of the carbon spheres obtained by hydrothermal carbonization of biomass are usually poor, which extensively limits the utilization of these carbon spheres as electrode materials [21,22]. Therefore, it is necessary to improve the surface properties and pore structures of these carbon spheres to obtain good electrode materials with excellent electrochemical performance [23,24]. It is well known that physical activation and chemical activation are the main activation methods for carbon materials. Among them, chemical activation is a very effective method due to its low activation temperature, short activation time, and high yield of carbon materials. Many reagents, such as KOH, ZnCl_2_, H_3_PO_4_, K_2_CO_3_, NaOH, etc., can be used as activators for chemical activation. Among these activators, KOH, ZnCl_2_, and H_3_PO_4_ are the most commonly used [25]. The carbon materials obtained from chemical activation contain a large number of micropores and mesopores, which is beneficial for improving the electrochemical performance [26,27]. Rey-Raap et al. explored the influence of carbon nanotubes addition in biomass-derived carbon materials, and a KOH-activated glucose-derived carbon material was prepared by hydrothermal carbonization of glucose in the presence of carbon nanotubes [28]. Results showed that the assembled supercapacitor had a high specific capacitance of 206 F/g at a current density of 0.2 A/g when only 2 wt% of carbon nanotubes was added. Fan et al. prepared a series of micro-mesoporous carbon spheres by hydrothermal carbonization of carrageenan and further activated by KOH at different temperatures, and a high specific capacitance of 230 F/g (current density 1 A/g) was achieved from the carbon spheres activated at 900 °C [29].

In this work, carbon spheres were prepared from the industrial hemicelluloses with high purity and uniform structures extracted from the dissolving pulp by hydrothermal carbonization [30]. Then, various activating reagents, such as KOH, K_2_CO_3_, Na_2_CO_3_, and ZnCl_2_, were used to activate the carbon spheres to improve the electrochemical performance of these hydrothermal carbon spheres as electrode materials. The structure and properties of these carbon spheres were investigated with scanning electron microscopy (SEM), X-ray photoelectron spectroscopy (XPS), N_2_ adsorption/desorption, and Raman spectroscopy. Furthermore, the activated carbon spheres were used as electrode materials in supercapacitors to evaluate their electrochemical performances.

## 2. Results and Discussion

### 2.1. Preparation of the Hydrothermal Carbon Spheres

Previous work showed that the hemicelluloses extracted from the dissolving pulp mainly comprised (1→4)-linked β-D-xylans, which is a good raw material for producing carbon spheres by hydrothermal carbonization [30]. In this work, the hydrothermal carbonization of the hemicelluloses was performed in 2% H_2_SO_4_ solution because the addition of acid could promote the degradation of hemicelluloses into furan compounds and significantly improve the yield of carbon spheres [15,31]. As compared with the carbon sphere yield of 33.9% obtained from water, a much higher carbon spheres yield of 44.6% was achieved when the hydrothermal carbonization was carried out in 2% H_2_SO_4_ solution. The morphology of the hydrothermal carbon spheres was detected with SEM and shown in Figure 1. The hydrothermal carbon spheres prepared in 2% H_2_SO_4_ solution were more regular and smooth.

### 2.2. Preparation and Characterization of the Activated Carbon Spheres

Generally, the carbon materials obtained from hydrothermal carbonization show poor specific surface area and porosity, which makes it difficult for them to meet the requirements of supercapacitors [15,21]. In this work, four activators, including KOH, K_2_CO_3_, ZnCl_2_, and Na_2_CO_3_, were used to improve the pore structure and specific surface area of the hydrothermal carbon spheres. The activated carbon spheres without the addition of the activator were recorded as ACS, and the carbon spheres activated with KOH, K_2_CO_3_, Na_2_CO_3_, and ZnCl_2_ were recorded as ACS-KOH, ACS-K_2_CO_3_, ACS-Na_2_CO_3_, and ACS-ZnCl_2_, respectively. Figure 2 shows the SEM images of these activated carbon spheres (ACS, ACS-KOH, ACS-K_2_CO_3_, ACS-Na_2_CO_3_, and ACS-ZnCl_2_). It can be seen that the surface of ACS was still smooth, and the morphology of ACS hardly changed during the calcination process. However, the surface of ACS-KOH, ACS-K_2_CO_3_, ACS-Na_2_CO_3_, and ACS-ZnCl_2_ became rough. Among them, the spherical shape of ACS-KOH was seriously damaged. Similar results were also observed in the KOH-activated carbon materials derived from glucose, and it was stated that the obvious etched morphology was correlated with the fusion of KOH drops on the surface or the formation of a liquid-phase intermediate during activation [28,32]. It is worth noting that although many pores were observed in the SEM images of ACS-KOH, these pores were mainly macropores. Mesopores and micropores can rarely be observed in SEM images because of their small sizes. Therefore, the mesopores and micropores of the samples were further characterized by N_2_ adsorption and desorption measurement.

The electrochemical performances of carbon materials are significantly affected by their porous structures. The specific surface area and pore volume of these carbon spheres are shown in Table 1. As compared with the ACS, the Brunauer-Emmett-Teller (BET)-specific surface area of these activated carbon spheres increased significantly, which was in the order of ACS-ZnCl_2_ > ACS-K_2_CO_3_ > ACS-KOH > ACS-Na_2_CO_3_. Similar trend was also observed from the pore volume results. Therefore, the porous structure of the carbon materials is closely related to the activators used. Among these activated carbon materials, the specific surface area of ACS-ZnCl_2_ (2025 m^2^/g) was much higher than others (441–1141 m^2^/g). The N_2_ adsorption/desorption isotherms of these carbon spheres are shown in Figure 3a. Typical type I isotherms were observed for all carbon sphere samples, and no obvious hysteresis loops were found. However, for ACS-K_2_CO_3_, ACS-KOH and ACS-Na_2_CO_3_ samples, the adsorption amount of N_2_ still gradually increased with the increment of P/P_0_ from 0.20 to 0.95, indicating that in addition to micropores, a small amount of mesopores existed in these activated carbon spheres. From the isotherm of ACS-ZnCl_2_, it was found that the adsorption amount of N_2_ significantly increased with the increment of P/P_0_ from 0.05 to 0.40 and then tended to be flat. It suggested that a relatively high content of mesopores but in small size was contained in the ACS-ZnCl_2_ sample [16,33,34]. Due to the difference of the activation mechanism, the porous structure of the carbon materials activated by different activators may have varied significantly. Heidarinejad et al. stated that potassium metal is thought to be introduced into the internal structure of the carbon matrix during the gasification process, leading to the expansion of existing pores and the creation of new ones [35]. For Zn ions, Rupar et al. reported that evaporation of carbo-thermally reduced Zn was known to be crucial in the zinc activation mechanism, as it caused the interlayer movement of zinc ions inside a carbon scaffold, which could further enhance pore development by internal tunnel formation [36]. Meanwhile, Heidarinejad et al. stated that ZnCl_2_ could enhance the condensed aromatic reactions by facilitating molecular hydrogen deformation from the hydro-aromatic structure of the precursors [35]. By increasing the amount of ZnCl_2_, more cracks may occur in the structure of activated carbon and result in the mesoporosity increase in the activated carbon structure, which further breaks down, and the micropores deform to form mesopores [35]. Moreover, the molecular size of the metallic salts intercalated in the carbon layers removed during the washing stage may also be responsible for their porosity and pore sizes [37,38]. Figure 3b shows the pore size distribution diagram of these carbon spheres calculated by the density functional theory (DFT) model. Results suggested that ACS presented a high ratio of micropores, while ACS-KOH, ACS-K_2_CO_3_, ACS-Na_2_CO_3_, and ACS-ZnCl_2_ also contained some smaller mesopores in addition to micropores. Notably, the pore sizes of ACS-ZnCl_2_ were mainly centered at 1–2 and 2–4 nm, indicating that ZnCl_2_ can significantly promote the formation of mesopores. The existences of these mesopores were conducive to the storage of charge and electrolyte, which can significantly accelerate the transport of electrolyte ions [39,40]. Therefore, these activated carbon spheres may have excellent electrochemical performances, especially ACS-ZnCl_2_.

The graphitization degrees of these activated carbon spheres were characterized with Raman spectra (Figure 3c). Two main peaks located at 1330 (D band) and 1590 cm^−1^ (G band) are related to the disordered carbon and the sp^2^-hybridized carbon atoms in the graphitic layers, respectively, and the ratio of D and G band intensities (I_D_/I_G_) is usually used to evaluate the degree of graphitization [41,42,43]. As compared with the I_D_/I_G_ value of ACS (0.96), the I_D_/I_G_ values of ACS-Na_2_CO_3_, ACS-KOH, ACS-K_2_CO_3_, and ACS-ZnCl_2_ changed slightly, which were 1.03, 0.97, 0.99, and 0.95, respectively. It suggested that the addition of activating reagents did not change the graphitization degrees of these carbon materials significantly under the given conditions.

The surface element compositions of these carbon spheres were further studied with XPS (Figure 3d–f). As shown in Figure 3d, only two peaks at 284.6 eV and 532.8 eV are observed in all samples, which are attributed to the C 1s and O 1s, respectively. The C and O contents of these carbon spheres obtained from the XPS are listed in Table 1. It was found that all these activated carbon spheres contained a large number of C atoms and a small amount of O atoms. As compared with the ACS, the C content of the carbon spheres decreased after activation, while the O content increased to some degree. The reason may be that the activators make more C atoms in the bulk to the surface of carbon materials, which tend to capture more O atoms to form functionalities [15]. The presence of oxygen-containing functional groups can improve the wetting ability of the carbon materials in the electrolytes. Meanwhile, some oxygen-containing functional groups, such as quinone or pyrone groups, may give rise to pseudocapacitance in the supercapacitor, which is beneficial for improving the electrochemical performance of carbon materials [12,44]. Figure 3e and f illustrate the C 1s and O 1s spectra of the ACS-ZnCl_2_, respectively. Four carbon bonds in C 1s spectrum: C–C/C=C (284.6 eV), C–O (286.0 eV), C=O (287.9 eV), and O–C=O (289.2 eV), and three different oxygen bonds in O 1s spectrum: C=O (531.3 eV), C–OH/C–O–C (532.6 eV), and –COOH (533.7 eV) were also observed [11,45].

### 2.3. Electrochemical Properties

The performance of ACS and these activated carbon spheres as electrode materials in a three-electrode system with 6 M KOH aqueous electrolyte was tested to estimate their potential as electrode materials. The electrochemical performances of ACS and these activated carbon spheres were investigated using cyclic voltammetry (CV) and GCD tests, and the corresponding results were presented in Figure 4. As can be seen from Figure 4a, the CV curves of ACS-KOH, ACS-K_2_CO_3_, ACS-ZnCl_2_, and ACS-Na_2_CO_3_ were quasi-rectangular at a scan rate of 50 mV/s between −1 and 0 V, implying the good capacitance behavior of these electrode materials. However, a triangle-like CV curve was observed for the ACS. The area under the CV curves decreased in the order of ACS-ZnCl_2_ ≈ ACS-K_2_CO_3_ > ACS-KOH > ACS-Na_2_CO_3_ > ACS. The specific capacitance of ACS-ZnCl_2_, ACS-K_2_CO_3_, ACS-KOH, ACS-Na_2_CO_3_, and ACS electrode from CV curves was 138, 137, 121, 95, and 71 F/g, respectively, suggesting that the ACS-ZnCl_2_ and ACS-K_2_CO_3_ electrodes possessed relatively high specific capacitance among all the activated carbon sphere electrodes. The GCD curves of these activated carbon spheres are shown in Figure 4b. As illustrated, a distorted triangle-like GCD curve with a huge IR drop was observed from the ACS sample, indicating the big resistance of the ACS sample. However, the GCD curves of the other activated carbon spheres were symmetrical and linearly triangular, and no obvious IR drop was found at the beginning of the discharge process, suggesting the favorable double-layer capacitance behavior and small resistance of these activated carbon spheres. The specific capacitances of these activated carbon spheres in a three-electrode system were calculated from the GCD curves using Equation (1) and shown in Table 1. The specific capacitance of ACS-ZnCl_2_ and ACS-K_2_CO_3_ electrode was 183 F/g and 172 F/g, respectively, at a current density of 1 A/g, which is much higher than that of the other samples. The orders of the specific capacitance values of these samples were in accordance with their specific surface areas. Additionally, it was found that although the specific surface area of ACS-ZnCl_2_ (2025 m^2^/g) was much higher than that of ACS-K_2_CO_3_ (1141 m^2^/g), the specific capacitances of them were very close. However, a much higher O content can be observed from the ACS-K_2_CO_3_ sample (13.45 at.%). Therefore, the specific capacitance of the carbon materials is not only related to the surface areas and porosity but also to their surface functionalities [46].

EIS tests were also carried out in the 10^−2^–10^5^ Hz frequency range to evaluate the electrochemical impedance of these activated carbon spheres. As shown in Appendix A, the near-vertical lines at low frequency represent the dominance of the EDLC, suggesting the good capacitive behavior of these activated carbon spheres. The intercept of the plot with real axis at high frequency is related to the equivalent series resistance (ESR), which includes the ionic resistance of electrolyte, the intrinsic resistance of active materials, and the contact resistance at the interface of active materials and the current collector [29]. A relatively small ESR can be observed for all the activated carbon spheres. The semicircle at the high frequency range is assigned to the charge-transfer resistance (*R*_ct_) from the interface of active materials and the electrolyte. Results showed that the *R*_ct_ of ACS-ZnCl_2_ is much higher than other activated carbon spheres. The high *R*_ct_ of ACS-ZnCl_2_ may be attributed to the poor wettability of the sample with lower O content. Previous works stated that the reasonable specific surface area, balanced pore distribution, appropriate functional groups, and the type of electrolyte could affect the conductivity and ion diffusion rate of the carbon electrode in the electrolyte and cause differences in electrochemical performance between different carbon materials [10]. Combining the results of specific surface area and pore volume of these carbon spheres, it is further confirmed that the existence of large specific surface area and pore volume is conducive to improving the electrochemical properties of carbon materials as electrode materials. The excellent performance of ACS-ZnCl_2_ can also be attributed to its developed micro/mesopores structure, since pore structure can facilitate the contact between the electrolyte and the carbon material. Based on the above results, ACS-ZnCl_2_ was further explored as an electrode material for the supercapacitor application.

To further optimize the electrochemical performance of ACS-ZnCl_2_, ZnCl_2_-activated carbon spheres were prepared under different ZnCl_2_/hydrothermal carbon spheres weight ratio (1, 2, 3, and 4). The CV and GCD results of these ACS-ZnCl_2_ in a three-electrode system were presented in Figure 5a,b, respectively. The results suggested that the specific capacitance of the ACS-ZnCl_2_ electrode increases with the dosage of ZnCl_2_. When the mass ratio of ZnCl_2_ to hydrothermal carbon spheres is 3:1, the specific capacitance is 183 F/g at 1A/g. However, the specific capacitance (188 F/g) is almost unchanged when the mass ratio of ZnCl_2_ to hydrothermal carbon spheres further increases to 4:1. Therefore, the performance of the ACS-ZnCl_2_ material obtained at a ZnCl_2_/hydrothermal carbon spheres weight ratio of 3:1 was tested to evaluate the potential of ACS-ZnCl_2_ as electrode materials in a three-electrode system with 6 M KOH as the electrolyte, and the results were shown in Figure 5c–e. The rate performance of ACS-ZnCl_2_ electrode was investigated using CV tests at different scan rates. Figure 5c shows that the CV curves exhibit a rectangular-like shape at all scan rates, and the area under the CV curves increases with the scan rates, increasing from 5 to 100 mV/s, suggesting the highly capacitive behavior of the prepared carbon materials with good ion response [47]. GCD curves of the ACS-ZnCl_2_ electrode at current densities from 0.2 to 10 A/g (Figure 5d) exhibited a slightly twisted isosceles triangle, and no obvious IR drop was observed at the beginning of the discharge process, indicating the good reversibility of the ACS-ZnCl_2_ electrode. The corresponding specific capacitance of the ACS-ZnCl_2_ electrode at different current densities is shown in Figure 5e. From the GCD curves, the specific capacitance of the ACS-ZnCl_2_ materials obtained at a ZnCl_2_/hydrothermal carbon spheres weight ratio of 3:1 was 218 F/g at a current density of 0.2 A/g. When the current density further increased to 10 A/g, a specific capacitance of 138 F/g was observed. The capacitive performance suggested a good rate capacitive behavior of the ACS-ZnCl_2_ materials [23].

The capacitive performance of the as-prepared ACS-ZnCl_2_-based symmetric supercapacitor was tested for practical application. The supercapacitor was assembled by employing two identical ACS-ZnCl_2_ electrodes in 2 M Li_2_SO_4_ electrolyte because of the high work potential of the neutral Li_2_SO_4_ electrolyte [16]. The CV curves of the ACS-ZnCl_2_-based supercapacitor in the voltage window of 0–1.8 V are shown in Figure 6a. The curve shape was almost unchanged with the increase in the scan rate from 10 to 300 mV/s, revealing a good rate performance of the ACS-ZnCl_2_-based supercapacitor. The GCD curves of the supercapacitor at current densities from 0.5 to 10 A/g are shown in Figure 6b. The IR drop is relatively low, suggesting that the supercapacitor has low internal resistance because of fast ion diffusion and charge transfer. The specific capacitances calculated from the GCD curves are summarized in Figure 6c. It can be seen that the specific capacity decreased with the increase in the current density. The specific capacitance is 137 F/g at 0.5 A/g and decreases to 100 F/g at 10 A/g, indicating a good rate capability of 73% capacitance retention with the growth of current densities from 0.5 to 10 A/g. Based on its high specific capacitance, a high energy density of 15.4 Wh/kg at 0.5A/g was obtained at a power density of 224 W/kg. The energy density of the ACS-ZnCl_2_-based symmetric supercapacitor is higher than that of conventional supercapacitors using activated carbon as the electrode material [5,47]. Moreover, the capacitance retention tests performed at 10 A/g for 2000 cycles are shown in Figure 6d. This suggested that more than 95% capacitance retention was observed after 2000 cycles.

## 3. Materials and Methods

### 3.1. Materials

The hemicelluloses extracted from the dissolving pulp were supplied by a viscose fiber mill in Xinjiang, China. The preparation of the hemicelluloses can be found in our previous literature [30]. Potassium carbonate (K_2_CO_3_, 99%), potassium hydroxide (KOH, 95%), sodium carbonate (Na_2_CO_3_, 99.5%), Zinc chloride (ZnCl_2_, 98%), Lithium sulfate (Li_2_SO_4_, 99%), polyvinylidene fluoride (PVDF), N-methyl pyrrolidone (NMP, 99%) were purchased from Shanghai Macklin Biochemical Co., Ltd., Shanghai, China. Acetylene black was obtained from Tianjin Ebory Chemical Co., Ltd., Tianjin, China. Hydrochloric acid (HCl, 37 wt%) and sulfuric acid (H_2_SO_4_, 95 wt%) were purchased from Shanghai Chemical Reagent Co., Ltd., Shanghai, China. HCl, H_2_SO_4_, KOH, K_2_CO_3_, Na_2_CO_3_, ZnCl_2_, Li_2_SO_4_, PVDF, NMP, and acetylene black were all of analytical grade and used as received.

### 3.2. Preparation of Carbon Spheres from the Hydrothermal Carbonization of Hemicelluloses

Hydrothermal carbonization was used to prepare the carbon spheres from the hemicelluloses. Typically, 3 g of hemicelluloses was placed into a Teflon-lined stainless-steel autoclave with 30 mL 2% H_2_SO_4_, and the autoclave was treated at 180 °C for 12 h. After hydrothermal treatment, the autoclave was cooled to room temperature. The mixture was filtered, and the cake was washed with ethanol and water until the filtrate was neutral, then dried at 80 °C for 12 h to obtain the hydrothermal carbon spheres.

### 3.3. Activation of the Hydrothermal Carbon Spheres with Different Activators

A certain amount of hydrothermal carbon spheres was treated with concentrated solutions of different activating reagents (KOH, K_2_CO_3_, Na_2_CO_3_, and ZnCl_2_) for 8 h at the desired mass ratio and then dried in an oven before calcination. A mass ratio of activator to carbon spheres of 3:1 was used, unless specified. The resulting mixture was calcined in a tube furnace at 800 °C for 2 h under the N_2_ flow. Finally, the char was thoroughly washed with 1 mol/L HCl solution and deionized water, and dried at 105 °C to obtain the activated carbon spheres. 

### 3.4. Characterization of the Prepared Carbon Spheres

The morphology of the carbon spheres was observed using scanning electron microscopy (SEM, JSM-6700F, JEOL Ltd., Tokyo, Japan). The specific surface areas and pore size distribution of carbon spheres were examined using the Micromeritics ASAP 2020 automatic analyzer by N_2_ adsorption and desorption of the isotherm at 77 K according to the Brunauer–Emmett–Teller (BET) method and the density functional theory (DFT) model, respectively, and the total pore volume was determined from the adsorption amount of N_2_ at a relative pressure of 0.99. Raman spectra of carbon spheres were collected using a LabRam Xplora confocal Raman microscope (Horiba Jobin Yvon, Longjumeau, France) according to a previous paper [48]. The surface chemical species of the carbon spheres were determined by X-ray photoelectron spectroscopy (XPS, ESCALAB 250Xi, Thermo Scientific, Waltham, MA, USA).

### 3.5. Electrochemical Measurements

To measure the electrochemical performance of the obtained carbon spheres, the working electrode of carbon spheres was prepared as follows. The carbon spheres were mixed with acetylene black and PVDF in a weight ratio of 8:1:1 in NMP to form slurries. The slurries were uniformly painted on a 1 cm^2^ area of nickel foam, pressed, and dried to obtain the electrode. The electrochemical tests of the electrode were carried out in a three-electrode system using a CHI760e electrochemical workstation (Shanghai Chen Hua Instruments Co., Shanghai, China) in 6 mol/L KOH aqueous solution. Hg/HgO electrode and Pt sheet were used as reference electrode and counter electrode, respectively. CV and GCD were investigated between −1 and 0 V, and the electrochemical impedance spectroscopy (EIS) was collected in a frequency range from 0.01 kHz to 100 kHz with 10 mV amplitude. Specific capacitance (*C*_1_, F/g) for the single electrode in a three-electrode system was calculated from the discharge curve using Equation (1).

The symmetric supercapacitor was fabricated using two pieces of working electrodes in a 2032-type coin cell with cellulose film as separator and 2 mol/L Li_2_SO_4_ solution as electrolyte. The preparation method of the electrode was described as above, and the mass of the carbon spheres loaded on each electrode was about 3 mg. The GCD tests were performed between 0 and 1.8 V at the current densities ranging from 0.5 to 10 A/g. Specific capacitance for the single electrode in the symmetric supercapacitor was calculated from the discharge curve using Equation (2), and the specific energy density (*E*, Wh/kg) and specific power density (*P*, W/kg) of the symmetric supercapacitor were calculated according to Equations (3) and (4), respectively.
*C*_1_ = *I* × ∆*t*/(*m* × ∆*V*)(1)
*C* = 2*I* × ∆*t*/(*m* × ∆*V*)(2)
*E* = *C* × (∆*V*)^2^/(8 × 3.6)(3)
*P* = 3600 × *E*/∆*t*(4)
where *I* (A) is the discharge current; ∆*t* (s) is the discharge time; *m* (g) is the mass of the carbon spheres loaded on each electrode; ∆*V* (V) represents the change of potential during discharge time ∆*t*.

## 4. Conclusions

This study demonstrated that carbon spheres with excellent electrochemical performance can be obtained from hemicelluloses by hydrothermal carbonization and chemical activation. Activation of these carbon spheres with KOH, K_2_CO_3_, Na_2_CO_3_, and ZnCl_2_ can significantly improve their specific surface area and pore volume. Among these activated carbon spheres, ACS-ZnCl_2_-based symmetric supercapacitor showed a high specific capacitance of 137 F/g at 0.5 A/g in 2 M Li_2_SO_4_ electrolyte and a large energy density of 15.4 Wh/kg at a power density of 224 W/kg. Moreover, the specific capacitance of the as-assembled symmetric supercapacitor can be retained at more than 95% of the initial specific capacitance at 10 A/g after 2000 cycles.

## Figures and Tables

**Figure 1 ijms-23-07101-f001:**
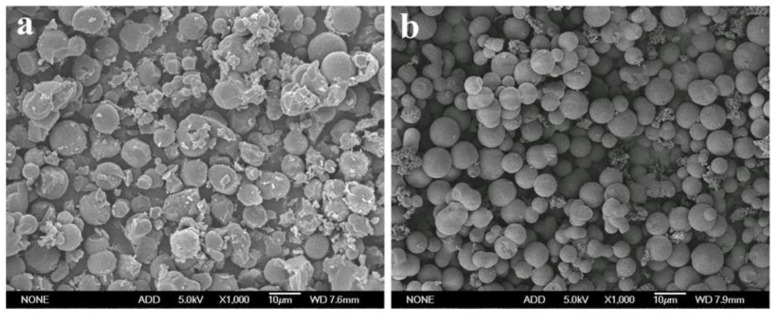
SEM images of the hydrothermal carbon spheres derived from hemicelluloses in (**a**) water and (**b**) 2% H_2_SO_4_ at 180 °C for 12 h at a magnification of 1000×.

**Figure 2 ijms-23-07101-f002:**
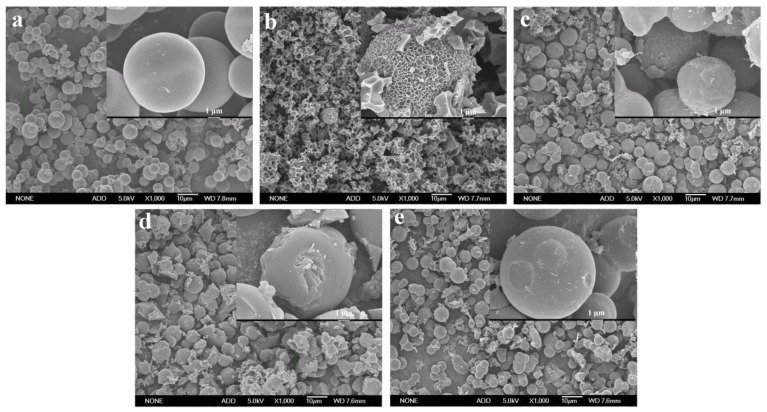
SEM images of (**a**) ACS, (**b**) ACS-KOH, (**c**) ACS-K_2_CO_3_, (**d**) ACS-ZnCl_2_, and (**e**) ACS-Na_2_CO_3_ at a magnification of 1000× and 10,000×, respectively.

**Figure 3 ijms-23-07101-f003:**
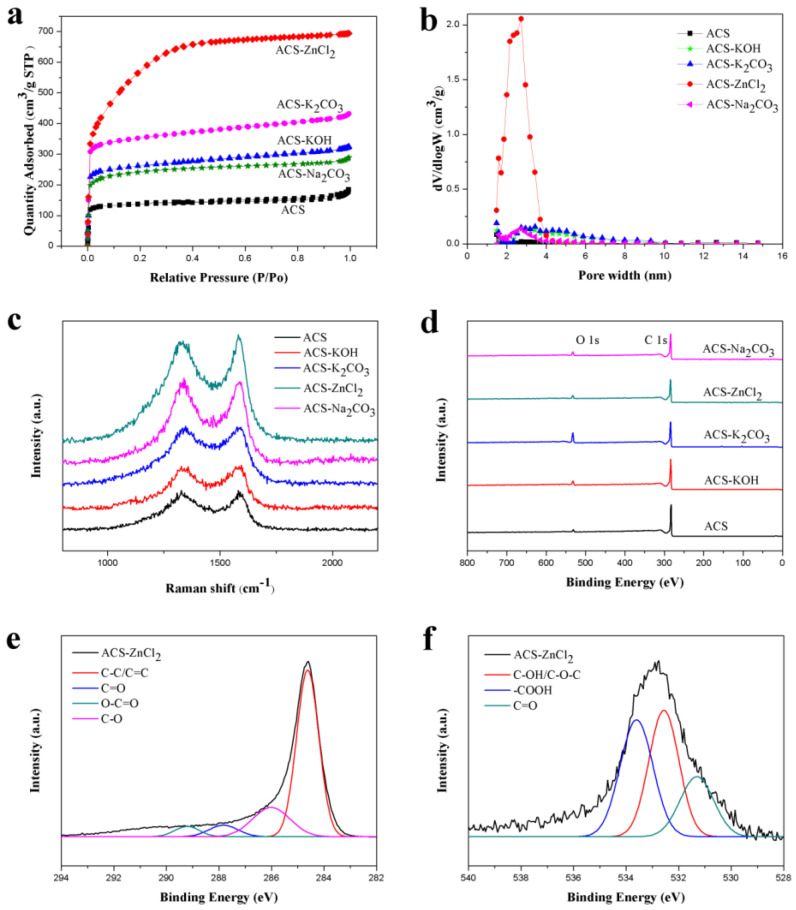
(**a**) N_2_ adsorption–desorption isotherms, (**b**) pore size distributions, (**c**) Raman spectra, (**d**) XPS spectra of activated carbon spheres, and XPS spectra of (**e**) C 1s and (**f**) O 1s of ACS-ZnCl_2_.

**Figure 4 ijms-23-07101-f004:**
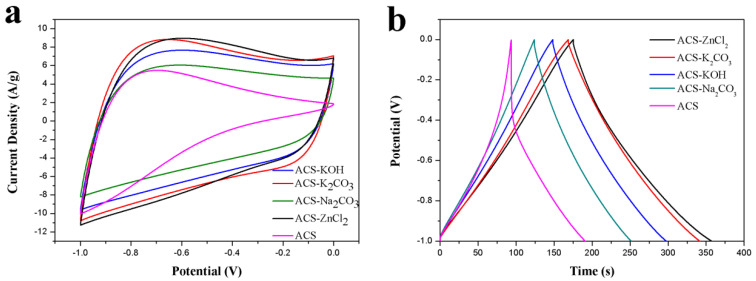
(**a**) CV curves (50 mV/s) and (**b**) GCD curves (1 A/g) of ACS, ACS-Na_2_CO_3_, ACS-KOH, ACS-K_2_CO_3_, and ACS-ZnCl_2_ in 6 M KOH in three-electrode system.

**Figure 5 ijms-23-07101-f005:**
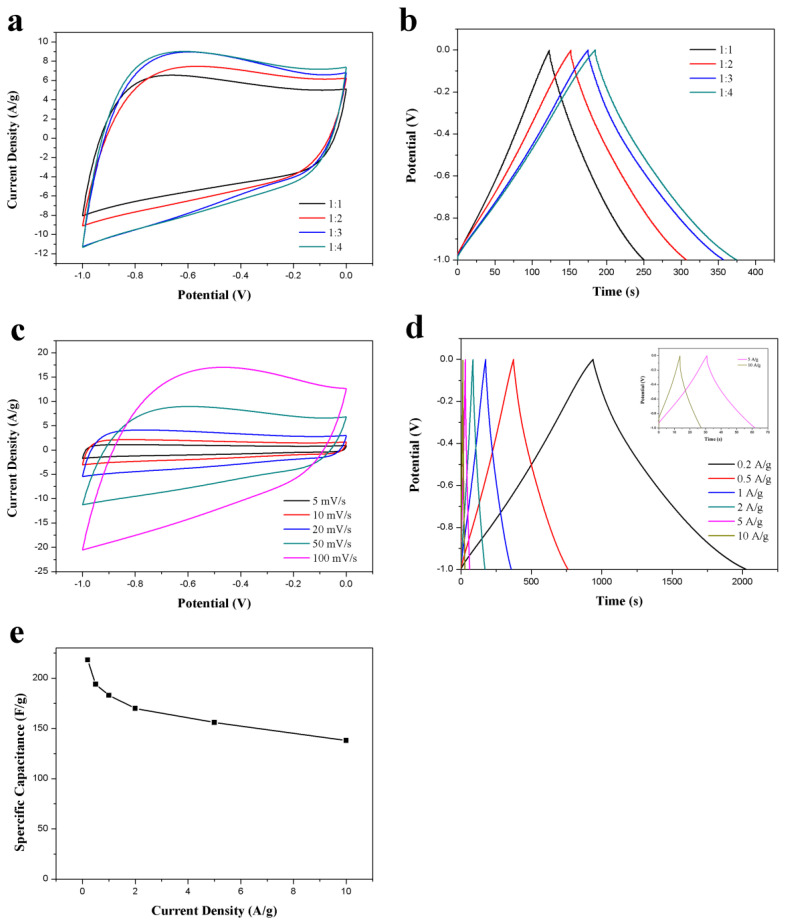
(**a**) CV curves (50 mV/s) and (**b**) GCD curves (1 A/g) of the activated samples obtained at different mass ratios of carbon spheres to ZnCl_2_; (**c**) CV curves of the ACS-ZnCl_2_ at different scan rates, (**d**) GCD curves and (**e**) specific capacitances of the ACS-ZnCl_2_ at different current densities. All measurements were performed in 6 M KOH in three-electrode system.

**Figure 6 ijms-23-07101-f006:**
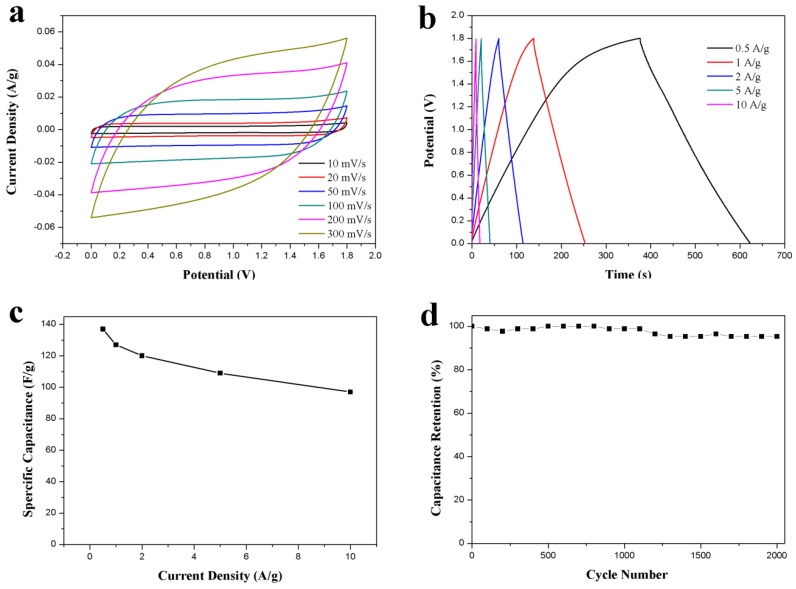
Electrochemical performance of the ACS-ZnCl_2_ in symmetric supercapacitor in 2 M Li_2_SO_4_ electrolyte. (**a**) CV curves tested at different scan rates, (**b**) GCD curves and (**c**) specific capacitances at different current densities, and (**d**) cycle stability for 2000 cycles at a current density of 10 A/g.

**Table 1 ijms-23-07101-t001:** The BET-specific surface areas, pore volume, surface element composition, and specific capacitance in three-electrode system of the carbon spheres activated with different activators.

Sample	ACS	ACS-KOH	ACS-K_2_CO_3_	ACS-Na_2_CO_3_	ACS-ZnCl_2_
BET-specific surface areas (m^2^/g)	441	852	1141	782	2025
Pore volume (cm^3^/g)	0.27	0.50	0.66	0.44	1.07
C (at.%)	96.27	93.44	86.55	93.68	94.11
O (at.%)	3.73	6.56	13.45	6.32	5.89
specific capacitance ^1^ (F/g)	98	149	172	128	183

^1^ The specific capacitance was obtained by galvanostatic charge–discharge (GCD).

## Data Availability

The data presented in this study are available on request from the corresponding authors.

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
