# Peer review of "Porous Carbon Spheres Derived from Hemicelluloses for Supercapacitor Application"

_ijms, 2022, doi:10.3390/ijms23137101_

Round 1

Reviewer 1 Report

Manuscript deals with activation of hemicelluloses for obtaining active materials for supercapacitors. Materials are properly characterized with relevant techniques for carbon materials, including BET, XPS and Raman. What is missing is maybe the conductivity measurement which are sometimes valuable, however this is mediated by use of active carbon in electrode preparation.

Electrochemistry is probed by CV and GCD and the calculations are properly done.

What is missing is maybe a discussion of the possible pore forming processes during activation.

References that might help with this:

10.1016/j.micromeso.2022.111790

10.1007/s10311-019-00955-0

Reviewer 2 Report

This manuscript reported a porous carbon sphere derived from hemicelluloses for supercapacitor application. The manuscript is clear for reading. Some concerns and suggestions are provided below:

1.     In SEM images, I found the developed porosity of ACS-KOH. However, why was values the BET specific surface area and pore volume of ACS-ZnCl2 and ACS-K2Co3 higher than those of ACS-KOH? Porosity is seen in the SEM images affects the BET specific area or pore volume. However, it seems so irrelevant in your manuscript. The correlation between analysis of SEM image, BET specific surface area, and pore volume are very important because of the effect on electrochemical properties. A sufficient discussion is required.

2.     In Raman spectra, I think that activating reagents didn’t influence the degree of graphitization and ID/IG is meaningless. You had better delete Raman spectra parts.

3.     In the line 219, where is Figure S1 for EIS tests?

4.     Why did ACS-ZnCl2 have the highest specific capacitance and energy density of samples?  Is it just because higher BET specific surface area or because the pores are developed? Please rewrite this part in more scientific approach, not just an interpretation.

Round 2

Reviewer 2 Report

The authors  answered  scientifically well for my review  and rewrote well their manuscript. So, I  accepted this paper for IJMS  publication.